# Ventilation-Associated Particulate Matter Is a Potential Reservoir of Multidrug-Resistant Organisms in Health Facilities

**DOI:** 10.3390/life11070639

**Published:** 2021-06-30

**Authors:** Evgenia Chezganova, Olga Efimova, Vera Sakharova, Anna Efimova, Sergey Sozinov, Anton Kutikhin, Zinfer Ismagilov, Elena Brusina

**Affiliations:** 1Department of Epidemiology, Kemerovo State Medical University, 650056 Kemerovo, Russia; chezganova@kemsma.ru; 2Institute of Coal Chemistry and Material Science, Siberian Branch of the Russian Academy of Sciences, 650099 Kemerovo, Russia; efimova@iccms.sbras.ru (O.E.); sozinov@iccms.sbras.ru (S.S.); ismagilov@iccms.sbras.ru (Z.I.); 3Department of Experimental Medicine, Research Institute for Complex Issues of Cardiovascular Diseases, 650002 Kemerovo, Russia; sahavm@kemcardio.ru (V.S.); kytiag@kemcardio.ru (A.K.); 4Kemerovo Regional Center for Hygiene and Epidemiology, 650002 Kemerovo, Russia; efimova@fguzko.ru

**Keywords:** particulate matter, air pollution, hospital dust, ventilation grilles, multidrug-resistant organisms, healthcare-associated infections, HCAI-causing pathogens, reservoir of infection, microbial diversity, chemical composition

## Abstract

Most healthcare-associated infections (HCAIs) develop due to the colonisation of patients and healthcare workers by multidrug-resistant organisms (MDRO). Here, we investigated whether the particulate matter from the ventilation systems (Vent-PM) of health facilities can harbour MDRO and other microbes, thereby acting as a potential reservoir of HCAIs. Dust samples collected in the ventilation grilles and adjacent air ducts underwent a detailed analysis of physicochemical properties and biodiversity. All Vent-PM samples included ultrafine PM capable of reaching the alveoli. Strikingly, >70% of Vent-PM samples were contaminated, mostly by viruses (>15%) or multidrug-resistant and biofilm-producing bacterial strains (60% and 48% of all bacteria-contaminated specimens, respectively). Total viable count at 1 m from the ventilation grilles was significantly increased after opening doors and windows, indicating an association between air flow and bacterial contamination. Both chemical and microbial compositions of Vent-PM considerably differed across surgical vs. non-surgical and intensive vs. elective care units and between health facilities located in coal and chemical districts. Reduced diversity among MDRO and increased prevalence ratio in multidrug-resistant to the total *Enterococcus* spp. in Vent-PM testified to the evolving antibiotic resistance. In conclusion, we suggest Vent-PM as a previously underestimated reservoir of HCAI-causing pathogens in the hospital environment.

## 1. Introduction

The rapid development and implementation of novel health technologies are intertwined with the growing complexities of the control for healthcare-associated infections (HCAIs). According to the recent estimates, HCAIs affect 3–15% of patients within inpatient health facilities [1,2] and these numbers are drastically higher in intensive care units, reaching an unacceptable 30% of the patients [3]. The total annual costs of HCAIs are around USD 10 billion in the United States [4] and GBP 3 billion in the United Kingdom [5]. In addition to the significant deterioration of the quality of life and prolonging the hospital stay, HCAIs are associated with an increased risk in adverse outcome, including death [6,7]. The burden of HCAIs is determined by multiple drug resistance (MDR), a phenomenon whereby certain bacterial strains survive at harsh hospital environment and acquire intricate mechanisms to become unresponsive to antibiotics, e.g., changes in the membrane structure limiting drug uptake; acquisition of the drug-resistant proteins by the horizontal transmission of the respective genes; structural alterations of the pharmacological targets; molecular barriers hindering drug access to the target, overexpression of the target proteins; production of the enzymes inactivating the drugs; and drug-specific efflux pumps [8,9,10,11,12]. Among the most frequent MDROs are *Escherichia coli*, *Staphylococcus aureus*, *Klebsiella* spp., *Acinetobacter baumannii*, *Pseudomonas aeruginosa*, *Enterococcus faecium*, *Streptococcus pneumoniae*, and coagulase-negative staphylococci [13,14,15].

The last decade has been highlighted by a broad implementation of minimally invasive surgical techniques, reduced duration of hospital stay, and the extension of applications of novel biomedical materials and devices. A growing amount of medical equipment within the operating rooms and intensive care units results in an increasing area of the surfaces which are potentially suitable for dust sedimentation, bacterial colonisation, and subsequent formation of cleaning-resistant and disinfection-resistant biofilms [16,17,18,19,20,21,22]. Recent studies demonstrated that virtually all common surfaces in healthcare settings contain dry biofilms covering microbial, mostly multi-resistant pathogens [23], and found that the profile of microorganisms in the air resembles that on the hospital surfaces [24]. Manipulations performed by the healthcare personnel contribute to the transfer of surface microorganisms into the hospital air [25,26]. The indicated risk, however, can be controlled through enhanced environmental cleaning and disinfection procedures [27]. Currently, hospital air quality is recognised as an important determinant in preventing airborne transmission of HCAIs [28] and ambient air pollution is a well-established risk factor of community-acquired pneumonia in both adults [29,30] and children [31,32,33]. The distributions and compositions of air streams as well as their microbial contamination are determined by the amount and velocity of natural ventilation [28,34,35]; implementation of sophisticated mechanical ventilation systems [28,35]; differential regulation of air pressure [35]; efficiency of air filtration [35]; indoor air temperature [35]; and relative humidity [35].

Both patient-derived bioaerosols [36] and environmental particulate matter (PM) [37,38] represent rapid and highly efficient route of bacterial [38,39] and viral [40,41] transmission. Different-sized PM fractions have distinct penetration depths: Coarse PM (PM_10_ or PM_2.5–10_ having aerodynamic diameter between 2.5 and 10 µm) are retained in the upper airways, while fine PM (PM_2.5_ or PM_0.1–2.5_ having aerodynamic diameter between 0.1 and 2.5 µm) reaches the alveoli, and ultrafine PM (PM_0,1_ having aerodynamic diameter ≤0.1 µm) might trespass the blood–air barrier and enter the circulation [42]. Accordingly, PM_0.1–2.5_ is predominant in the lungs and is mainly responsible for pneumonia [29,30] and chronic obstructive pulmonary disease [43,44,45], while PM_2.5–10_ is primarily associated with upper respiratory tract infections [46,47]. Despite the well-defined role of PM in respiratory diseases, the role of hospital PM as a vehicle for bacterial and viral agents causing HCAIs has been largely neglected to date, albeit the critical reviews suggested the potential importance of ventilation system contamination in the spread of HCAIs [48,49].

Here, we combined a materials science approach with a microbiological investigation to analyse the PM collected from the ventilation grilles and adjacent air ducts (Vent-PM) of various health facilities. We compared the physicochemical features and biodiversity in Vent-PM obtained from: (1) non-surgical and surgical units as surgical site infections are responsible for the majority (>40%) of HCAIs [13]; (2) elective and intensive care units since HCAIs are the most prevalent in the latter setting compared to the others [3]; (3) health facilities located in urban areas exclusively possessing coal mines or chemical plants, as these enterprises generate distinct types of PM [50,51].

## 2. Materials and Methods

### 2.1. Sampling and PM Chemical Evaluation

Samples of hospital Vent-PM (*n* = 128) were collected from the local exhaust ventilation systems (ventilation grilles and adjacent ductwork) into the sterile plastic containers using the sterile gloves and were further compared between the following: (1) non-surgical (*n* = 88) and surgical (*n* = 40) units; (2) elective (*n* = 80) and intensive care (*n* = 48) units; (3) health facilities located within the urban areas containing exclusively coal (*n* = 27) or chemical (*n* = 36) enterprises. The distribution of Vent-PM samples among the study groups is indicated in Table 1, which shows the distribution across all comparisons, and Table 2, which demonstrates the distribution specifically across industrial districts.

The entire metadata for all collected samples is represented in Table 3.

For visualisation, Vent-PM samples (*n* = 16 for non-surgical and *n* = 12 for surgical units; *n* = 18 for elective care and *n* = 10 for intensive care units; *n* = 11 for coal and *n* = 17 for chemical districts) were mounted on a double-sided adhesive conductive carbon tape on the aluminum stubs and visualised by scanning electron microscopy (JSM-6390LA, JEOL, Tokyo, Japan) at 30 kV accelerating voltage and 1 nA probe current. Elemental analysis was carried out by energy-dispersive X-ray spectroscopy (JED-2300, JEOL, Tokyo, Japan) at 133 eV spectral resolution. For each Vent-PM sample, we defined three representative quadrants and measured the average atomic percent for each element employing the ZAF correction method and then we calculated the weight percent of each element. CHNSO analysis was performed in the same Vent-PM samples (*n* = 28) by a catalytic oxidation of the particles at 1060 °C (Flash 2000, Thermo Fisher Scientific, Waltham, MA, USA).

A particle-size distribution curve of hospital Vent-PM (*n* = 16 for non-surgical and *n* = 12 for surgical units; *n* = 18 for elective care and *n* = 10 for intensive care units; *n* = 11 for coal and *n* = 17 for chemical districts) was assessed by dynamic light scattering (Zetasizer Nano ZS, Malvern Instruments, Malvern, UK). Before the measurement, Vent-PM samples were resuspended in sterile-filtered double distilled water, ultrasonicated for 20 min to disaggregate the particles, sequentially filtered through the 1.0, 0.45, and 0.22 µm pores to separate the ultrafine PM, and then incubated at 25 °C for 20 min. All measurements were performed five times (50 runs per measurement) with the further calculation of the average distribution.

### 2.2. Viruses Evaluation

Detection of the RNA belonging to group A rotaviruses (Rotavirus A), genotype II noroviruses (Norovirus genogroup II), astroviruses (Astrovirus), enteroviruses (Enterovirus), hepatitis A virus and SARS-CoV-2, as well as DNA of *Shigella* spp., enteroinvasive *Escherichia coli*, *Salmonella* spp., and thermophilic *Campylobacter* spp. was performed by real-time polymerase chain reaction (AmpliSense-Rotavirus/Norovirus/Astrovirus-FL (V40 (RG,iQ,FEP), Central Research Institute of Epidemiology, Moscow, Russia), AmpliSense-Enterovirus-FL (R-V16-F, Central Research Institute of Epidemiology, Moscow, Russia), AmpliSense-HAV-FL (V4-FEP, Central Research Institute of Epidemiology, Moscow, Russia), RealBest RNA SARS-CoV-2 (D-5580, Vector-Best, Novosibirsk, Russia), AmpliSense-*Shigella* spp.-FL/AmpliSense-EIEC/*Salmonella* spp./*Campylobacter* spp.-FL (B44 (RG,iQ,FEP), Central Research Institute of Epidemiology, Moscow, Russia)) in all Vent-PM samples (*n* = 128).

### 2.3. Bacterial Evaluation

In order to investigate the bacterial diversity, all Vent-PM samples (*n* = 128) were seeded into the dextrose broth (M044, HiMedia Laboratories, Mumbai, India) and incubated at 37 °C for 24 h. Next, inoculates were re-seeded onto the blood agar (CM0055B, Thermo Scientific, Waltham, MA, USA), CHROMagar Candida (CA223-25, CHROMagar, Paris, France), or CHROMagar Orientation (RT413-25, CHROMagar, Paris, France) for 24 h at 37 °C. In order to obtain the pure culture, inoculates were re-seeded onto the Kligler’s iron agar (O16, Research Institute for Applied Microbiology and Biotechnology, Obolensk, Russia) with the following incubation at 37 °C before colonies became clearly distinguishable. Molecular typing was conducted using the VITEK^®^ 2 COMPACT Microbial Detection System (BioMerieux, Marcy-l’Etoile, France). Briefly, suspension cultures with the optical density of 0.50–0.65 McFarland turbidity standards were loaded into either VITEK^®^ 2GN (21341, BioMerieux, Marcy-l’Etoile, France) or VITEK^®^ 2GP identification cards (21342, BioMerieux, Marcy-l’Etoile, France) for the detection of Gram negative and Gram positive species, respectively, according to the manufacturer’s instructions. Upon 5–10 h, an automated algorithm compared the biochemical profile of the microorganisms with the reference standards in the database. Antibiotic resistance was determined in a similar manner (*n* = 44 Vent-PM samples) utilising VITEK^®^ 2 AST cards (BioMerieux, Marcy-l’Etoile, France) according to the manufacturer’s instructions.

The presence of biofilms at the surfaces of ventilation grilles (*n* = 29) and other hospital surfaces (*n* = 121) was examined by measuring the activity of a bacterial catalase using H_2_O_2_ containing films (BFR peroxyfilm, BFR Laboratories, Moscow, Russia). Total viable count (colony-forming units per m^3^ air) was quantified in 0.25 m^3^ air, which was aspirated by the impactor air sampler (Flora-100, Central Research Institute of Epidemiology, Moscow, Russia) at ≈1 m distance from the ventilation grilles (*n* = 19) before and after opening doors and windows.

### 2.4. Fungi Evaluation

Vent-PM samples were resuspended in Sabouraud dextrose broth (M033, HiMedia Laboratories, Mumbai, India), incubated for 24 h at 35 °C, then inoculated into Sabouraud dextrose agar (M063, HiMedia Laboratories, Mumbai, India) and CHROMagar Candida (CA223-25, CHROMagar, Paris, France), and finally incubated for 7 days at 35 °C. Identification of pathogenic yeasts was carried out by VITEK^®^ 2 YST identification cards (21343, BioMerieux, Marcy-l’Etoile, France) employing VITEK^®^ 2 COMPACT Microbial Detection System (BioMerieux, Marcy-l’Etoile, France). Molds were detected by a microscopic examination.

### 2.5. Statistical Analysis

Statistical analysis was carried out by GraphPad Prism 8 (GraphPad Software, San Diego, CA, USA). Descriptive data were represented by the median with interquartile range. Unpaired (independent) and paired (before-after) groups were compared by Mann–Whitney *U*-test and Wilcoxon matched-pairs signed rank test, respectively. *P* values ≤ 0.05 were defined as statistically significant.

## 3. Results

### 3.1. Chemical Composition of Hospital Vent-PM Is Environment-Dependent and Differs between Non-Surgical and Surgical Units but Not Elective and Intensive Care Units

We first examined the appearance of hospital Vent-PM, revealing two main geometric patterns: globular particles (13/28 samples, 46.4%, Figure 1A) and microscale fibres (15/28 samples, 53.6%, Figure 1B), which, however, did not correlate with the unit specialisation (Figure 2A), health facility location (Figure 2B), or bacterial (Figure 2C) or viral (Figure 2D) contamination. Measurement of particle-size distribution by means of dynamic light scattering found ultrafine PM in all samples (Figure 3). No considerable differences have been documented between surgical and non-surgical units (Figure 3A) as well as intensive and elective care units (Figure 3B), yet the particle-size distribution significantly differed between Vent-PM samples collected in health facilities located within coal and chemical districts (Figure 3C).

Accordingly, Vent-PM sampled in distinct units had almost similar mineral compositions consisting of carbon, oxygen, nitrogen, hydrogen, magnesium, sodium, aluminum, silicon, phosphorus, sulfur, chlorine, potassium, calcium, and iron. Intriguingly, higher amounts of nitrogen were detected in Vent-PM from non-surgical vs. surgical units and wards vs. operating rooms (Figure 4A). Vent-PM from the health facilities within the coal districts contained higher proportion of carbon, oxygen, and hydrogen but lower content of potassium than compared with those from the chemical districts (Figure 4B). Virus-contaminated Vent-PM had higher proportion of nitrogen but lower proportion of oxygen (Figure 4C). The composition of contaminated Vent-PM, however, differed between non-surgical and surgical units, as the former had higher proportions of carbon and nitrogen (Figure 4D). Mineral composition of Vent-PM collected in elective and intensive care units did not differ significantly (data not shown).

### 3.2. Hospital Vent-PM Is Frequently Contaminated by Multidrug-Resistant Organisms and Viruses

Microbiological analyses identified the microorganisms in 90/128 (70.3%) Vent-PM samples. Bacteria, viruses and fungi were detected in 81/128 (63.3%), 20/128 (15.6%), and 12/128 (9.4%) Vent-PM samples, respectively; 16/128 (12.5%) samples were contaminated by both bacterial and viral pathogens (Figure 5A). Hospital Vent-PM was notable for the high microbial diversity. In particular, bacteria were represented by 23 genera (Figure 5B) and Gram negative microbes prevailed over Gram positive microbes (62.5 and 37.5%, respectively). Molecular typing revealed the presence of *Acinetobacter baumannii*, *Acinetobacter haemolyticus*, *Acinetobacter lwoffi, Aeromonas sobria, Aeromonas salmonicida, Brevundimonas diminuta, Bordetella bronchiseptica, Campylobacter* spp., *Chromobacterium violaceum*, *Cronobacter dublinensis, Enterococcus faecium*, *Enterococcus faecalis*, *Enterococcus durans*, *Enterococcus gallinarum, Klebsiella pneumoniae*, *Kluyvera intermedia, Moraxella lacunata*, *Micrococcus* spp., *Pantoea* spp., *Pasteurella canis*, *Pasteurella testudinis, Pseudomonas aeruginosa, Pseudomonas luteola*, *Rhizobium radiobacter, Roseomonas gilardii, Raoultella ornithinolytica*, *Sphingomonas paucimobilis*, *Shewanella putrefaciens*, *Serratia plymuthica*, *Salmonella* spp., *Staphylococcus pseudintermedius*, *Staphylococcus hominis ssp. hominis*, and *Staphylococcus faecalis.* Sapronotic agents were responsible for 56.2% of all bacterial diversity, with *Sphingomonas paucimobilis* (9.4%), *Micrococcus* spp. (9.4%), and *Acinetobacter* spp. (6.3%) being the most frequent. Around 26% of bacteria were undefined because of their metabolic dormancy in the spore form. The total viable count near (≈1 m) ventilation grilles was significantly increased after opening doors and windows (from 163 to 303 colony-forming units per m^3^ air, *p* = 0.026, Appendix A), which indicates an association between the circulation of air flow and ventilation-associated bacterial contamination.

Antibiotic resistance profiling of 44 Vent-PM samples revealed that ≈60% (26/44) of them harboured MDRO (Figure 5C). In order to measure the bacterial evolution in the direction of antibiotic resistance, we calculated the prevalence ratio of multidrug-resistant to total *Enterococcus* spp. (the most frequent bacterial genus in Vent-PM) and found it ≈1.77 (38.7% to 21.9%, Figure 5C). Taken together, with reduced diversity amongst MDRO (13 genera, Figure 5C) as compared to the entire bacterial community in the Vent-PM (23 genera, Figure 5B), this suggested the Vent-PM as a potential reservoir of evolving antibiotic resistance.

Viral diversity was limited to the Rotavirus (56.5% of virus-contaminated samples), Norovirus (13.0%), and Betacoronavirus (30.4%) genera (Figure 5D). Out of 35 Vent-PM samples collected in health facilities with officially documented cases of COVID-19, seven (20.0%) contained SARS-CoV-2 RNA, indicating Vent-PM as a potential vehicle for its airborne transmission. Among the fungal species, we found molds but not pathogenic yeasts in all 12 contaminated samples. None of the identified fungi belonged to *Aspergillus* spp.

The detailed information for each sample is summarised in Table 4.

### 3.3. Microbial Composition of Hospital Vent-PM Is Patient-Dependent and Environment-Dependent

Vent-PM from the surgical units were most frequently contaminated by bacteria (32/40 samples, 80.0%), with only a minor proportion of MDRO (7/40, 17.5%) and rare (1/40, 2.5%) contamination by viruses or fungi (Figure 6A). Most of these samples were contaminated by Gram negative microorganisms (88.2%), while only one-third contained Gram positive species (33.3%). In contrast, non-surgical units were less frequently contaminated (58/88 samples, 65.9%, *p* = 0.16) but were characterised by a significantly higher proportion of virus-infected and fungi-infected Vent-PM samples (19/88, 21.6% and 11/88, 12.5%, *p* = 0.004, respectively, Figure 6A) and by the predominance of Gram negative species (53.1%) over Gram positive (46.9%). However, the prevalence of MDRO in Vent-PM did not differ between surgical and non-surgical units (7/40, 17.5% and 19/88, 21.6%, *p* = 0.77, Figure 6A).

Bacterial diversity in Vent-PM from non-surgical and surgical units was represented by 14 and 19 genera, respectively (Figure 6B). *Sphingomonas* spp. and *Pantoea* spp. prevailed in Vent-PM from surgical units (≈30% taken together), whereas *Enterococcus* spp. and *Micrococcus* spp. were responsible for >40% bacteria in Vent-PM from non-surgical units (Figure 6B). Vent-PM from surgical units harboured seven genera of MDRO without a clearly defined leading strain, in contradistinction to non-surgical units where *Enterococcus* spp. contributed to 50% of all MDRO (Figure 6C). Along with the prevalence ratio of multidrug-resistant to total *Enterococcus* spp. ≈1.81 (50.0% to 27.7%), this pointed at an ongoing evolution towards the development of a predominant antibiotic-resistant clone in non-surgical units (Figure 6C). Among the viruses, Rotavirus was the most frequent genus in Vent-PM (≈55% of all viruses, Figure 6D).

Regarding the comparison between intensive and elective care units, no differences in prevalence of contaminated Vent-PM samples have been revealed (32/48, 66.7% and 58/80, 72.5%, respectively, *p* = 0.62, Figure 7A). Nevertheless, Vent-PM from intensive care units showed considerably lower rates of bacterial contamination than Vent-PM collected in elective care units (24/48, 50.0% and 58/80, 72.5%, respectively, *p* = 0.017) but higher frequency of fungal contamination (9/48, 18.7% vs. 3/80, 3.7%, *p* = 0.012) (Figure 7A). The ratio of Gram positive and Gram negative bacteria also differed between intensive and elective care units (61.9% Gram positive vs. 38.1% Gram negative in the former and 76.1% Gram positive vs. 23.8% Gram negative in the latter setting). Importantly, despite the lower bacterial contamination, Vent-PM from the intensive care units demonstrated higher colonisation by MDRO than samples from elective care units, albeit this did not reach statistical significance (13/48, 27.1% vs. 13/80, 16.3%, respectively, *p* = 0.21, Figure 7A).

Vent-PM collected in the intensive care units was featured by the preponderance of *Enterococcus* spp. (≈43% of all bacteria, Figure 7B), which also composed the majority of MDRO (≈53%, Figure 7C). Albeit Vent-PM from elective care units was populated by a variety of bacterial species (Figure 7B), four genera contributed to almost two-thirds of MDRO (*Enterococcus* spp., *Klebsiella* spp., *Sphingomonas* spp. and *Bordetella* spp., Figure 7C). The prevalence ratio of multidrug-resistant to total *Enterococcus* spp. was ≈1.24 (53.3% to 42.9%) in intensive care and ≈2.63 (25.0% to 9.5%) in elective care units (Figure 7B,C), indicating a higher rate of antibiotic resistance development in the latter. Most probably due to the COVID-19 pandemic, Betacoronavirus was the main viral genus in Vent-PM from intensive care units (≈80% of all viruses), while Rotavirus was the leading taxon in the samples from elective care units (70%) (Figure 7D).

Strikingly, Vent-PM was contaminated in 88.9% (24/27) of health facilities located in coal districts; bacterial species were detected in all of these samples, while viruses and fungi were less often encountered (4/27, 11.1%, Figure 8A). Vent-PM samples from 36 health facilities located in chemical districts were contaminated in only 21 (58.3%) cases (*p* = 0.017), harbouring bacteria in 18/36 (50.0%) and viruses in 9/36 (25.0%) specimens (Figure 8A). The prevalence of MDRO in health facilities did not differ between coal and chemical districts (*p* = 0.90, Figure 8A).

Bacterial diversity in Vent-PM from coal district-associated health facilities was also higher whereas ≈46% of microbes found in Vent-PM from health facilities located in chemical districts were related to *Enterococcus* spp. and *Micrococcus* spp. (Figure 8B). In both locations of health facilities, the diversity of MDRO was almost twofold lower (7/8 genera) as compared to the total bacterial diversity (15/14 genera) (Figure 8B,C). We documented a predominance of multidrug-resistant *Enterococcus* spp. in Vent-PM from health facilities located in chemical districts, whereas none of the multidrug-resistant strains prevailed in coal districts (Figure 8C). Similar to the non-surgical and elective care units, the prevalence ratio of multidrug-resistant to total *Enterococcus* spp. in health facilities located in chemical districts was ≈1.75 (57.1% to 32.6%), which suggests the evolution towards antibiotic resistance (Figure 8B,C). The most of viral strains belonged to the Rotavirus genus (Figure 8D).

Employing the peroxide-based biofilm detection (where the bubbles appearing at the surface as a result of the interaction between the peroxide detector and bacterial catalase indicate a positive reaction), we revealed biofilms at 14/29 (48.3%) ventilation grilles and 11/121 (9.1%) other hospital surfaces (*p* = 0.0001, Figure 9). Analysis of Vent-PM from the indicated 14 biofilm-positive ventilation grilles found a high rate (6/14, 42.9%) of MDRO detection, suggesting Vent-PM as a potential reservoir for biofilm-producing strains, which are particularly hazardous in terms of multidrug resistance. Among the detected MDRO were *Roseomonas gelardii*, *Serratia plymutica*, *Sphingomonas pacimobilis*, *Enterococcus faecium*, and *Klebsiella pneumoniae* (2).

## 4. Discussion

Hospital environment represents a unique ecosystem with a rapidly ongoing molecular evolution, which favors the selection and spread of MDRO [52,53,54] that is an immense burden for both healthcare and economy responsible for around 700,000 annual deaths worldwide [55]. If not counteracted by antimicrobial stewardship, bacterial evolution might eventually result in the formation of a “superbug” resistance to the routine antibiotics, spreading across the distinct habitats within the health facilities, superseding other strains and causing HCAIs [56,57,58,59]. In the majority of cases, such MDRO evolve on various hospital surfaces because, in spite of their availability for regular cleaning and disinfection [7,60,61], these measures have limited efficiency and significantly depends on the compliance of medical staff [62,63]. Furthermore, recent studies indicated the potential importance of distant sources in harbouring MDRO [64]. Ventilation grilles and adjacent air ducts, which in many cases are rarely cleaned and are often hard-to-reach for the proper disinfection, have been suggested as remote shelters for MDRO [49,54], yet the respective evidence is merely anecdotal [65]. However, as PM is well-known for carrying pathogenic bacteria [38,39] and viruses [40,41], we hypothesised that the dust from the hospital ventilation systems might represent a potential vehicle for multiple MDRO. As distinct hospital units have different compositions of microbial populations [66,67] and surgical site infections account for the significant proportion of HCAIs [13], we compared surgical and non-surgical units in relation to the physicochemical and microbial profile of Vent-PM. Furthermore, we made a similar comparison between intensive and elective care units by taking into account a substantially higher prevalence of HCAIs in the former [3]. With respect to the earlier reports on the correlations between outdoor and indoor air pollution [68,69], we investigated whether the features of ambient air pollution are associated with the respective changes in hospital Vent-PM.

We found that hospital Vent-PM is heterogeneous and largely defined by both outdoor and indoor environment, as health facilities located within coal and chemical districts as well as non-surgical and surgical units were characterised by distinct elemental composition of Vent-PM. The presence of ultrafine PM in all Vent-PM samples confirmed the possibility of its circulation in the hospital air and penetration into the patient alveoli. Furthermore, Vent-PM was notable for the high rate of microbial (>70%) and particularly bacterial (>63%) contamination. Analysis of bacterial diversity indicated 23 genera associated with Vent-PM and a remarkable abundance of MDRO (≈60% of bacteria-contaminated samples), which generally had reduced diversity as calculated by multidrug-resistant to total *Enterococcus* spp. ratio. Vent-PM from the surgical units showed higher bacterial but lower viral and fungal contamination, although without a trend to the increase in MDRO proportion. Conversely, Vent-PM collected in intensive care units was less frequently colonised by bacteria but more often contained fungi in comparison with elective care units, also without statistically significant differences regarding MDRO. As pathogenic yeasts (including *Candida* spp.) have not been detected, fungal diversity was limited to molds. Unexpectedly, microscopic examination also did not reveal *Aspergillus* spp. in Vent-PM samples. Possible explanations include extremely cold winters in Siberia (−45 °C in January) in combination with low air humidity (≈30%) in healthcare facilities due to the central heating system working ≈8 months annually.

The contamination of Vent-PM from intensive care units by Betacoronavirus RNA demonstrated the dependence of Vent-PM microbial composition on patient setting, as some of these units specialised on the treatment of severe COVID-19. Moreover, it suggested a plausible role of Vent-PM in Betacoronavirus transmission, as SARS-CoV-2 can be detected even at ≈4 m from the patients [70,71,72]. In contrast, Vent-PM samples from elective care units were devoid of Betacoronavirus RNA, instead being contaminated with Rotavirus and Norovirus. In addition to the treatment specialisation, the microbial composition of Vent-PM was significantly impacted by the environment, as Vent-PM sampled in the health facilities from coal districts was 1.5-fold more often contaminated than Vent-PM from medical organisations located in chemical districts.

Taken together, our findings suggest dynamic interactions between the environment, patients, and Vent-PM acting as a harbourer for MDRO. Reduced biodiversity among MDRO compared to the total bacterial community and elevated prevalence ratio of multidrug-resistant to total *Enterococcus* spp. in Vent-PM testified to the evolving antibiotic resistance in the hospital ventilation.

Hospital surfaces are well-known for sheltering microbial pathogens by supplying them with a physical scaffold for the formation of biofilms, which is a self-produced molecular barrier protecting the microorganisms from cleaning agents and disinfectants and therefore creating a putative reservoir of antimicrobial resistance [23]. As the profile of microorganisms on the hospital surfaces is reminiscent of those circulating in ambient air [24], we assessed the biofilm-forming ability of the bacteria settling the ventilation grilles as compared with those colonising other hospital surfaces. We found a notably increased prevalence in biofilms on ventilation grilles as compared with other hospital surfaces. A substantial proportion of these biofilms (≈43%) harboured MDRO. Probably, irregular or inefficient cleaning of ventilation grilles promoted the survival of biofilm-producing and often multidrug-resistant bacteria, eventually resulting in the formation of biofilms on the ventilation-associated surfaces and also Vent-PM. These results underscore the potential importance of Vent-PM as a reservoir of multidrug resistance and underlines its relevance to the development of HCAIs.

Our results highlight the significance of Vent-PM for sheltering pathogenic microbes and indicate ventilation grilles together with the adjoining ductwork as a habitat for MDRO. The regular cleaning and disinfection of air ducts, possibly by the modern no-touch devices [60,73], might therefore contribute to the prevention of HCAIs. However, an impact of antimicrobials on the molecular evolution of micro-organisms colonising Vent-PM is unclear, particularly in relation to the bacteria emerging from the surfaces available for regular cleaning, which are frequently exposed to disinfectants. Further studies in this direction, including those employing next-generation sequencing, are required to better uncover the diversity and survivability of MDRO and other microbes in Vent-PM. Relatively high frequencies of viral contamination of Vent-PM points at the respective mechanism of airborne disease transmission, although the relative impact of bacterial and viral contamination of Vent-PM on the development of HCAIs has yet to be defined. However, viral transmission through Vent-PM should not be neglected in light of the COVID-19 pandemic.

## 5. Conclusions

PM from the hospital ventilation systems harbour a wide, patient-dependent, and environment-dependent spectrum of bacterial and viral pathogens and may be considered as a reservoir for biofilm-producing MDRO, which is a main culprit of HCAIs.

## Figures and Tables

**Figure 1 life-11-00639-f001:**
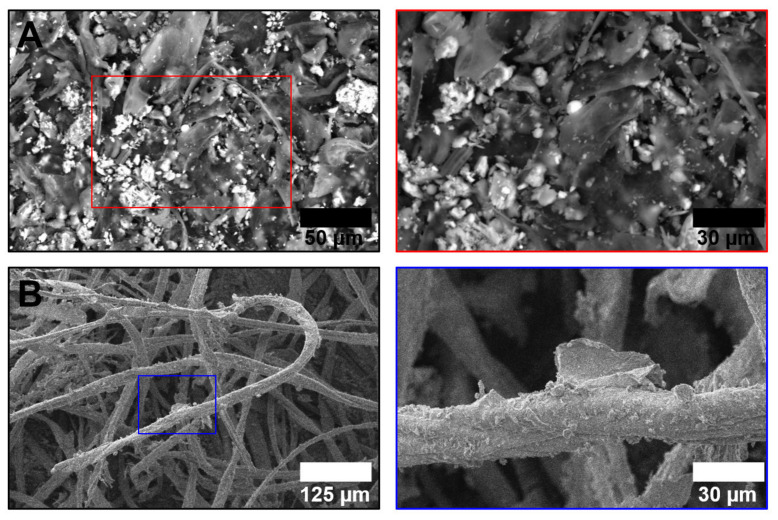
Structural features of hospital Vent-PM. (**A**) Globular particles. Left image: ×500 magnification. Right image: ×835 magnification. (**B**) Microscale fibres. Left image: ×200 magnification. Right image: ×835 magnification. Right images represent the close-ups of the left. Red and blue squares demarcate close-ups for globular particles and microscale fibres, respectively. All indicated images were acquired at 30 kV accelerating voltage and 1 nA probe current.

**Figure 2 life-11-00639-f002:**
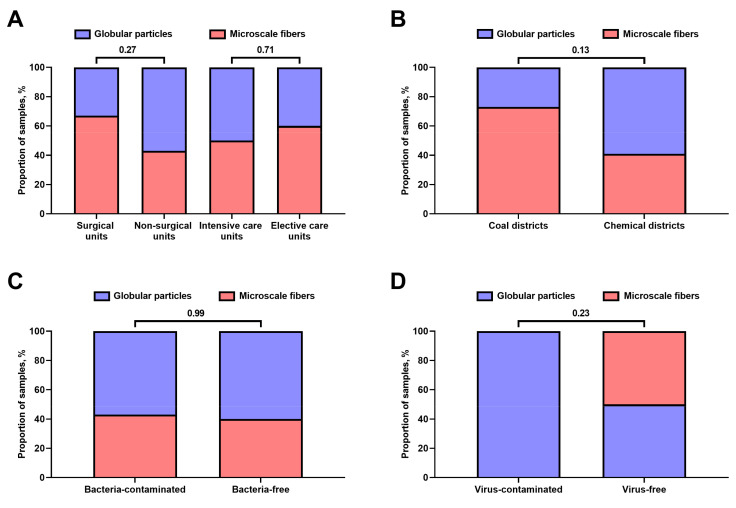
Distribution of distinct Vent-PM appearances in the following: (**A**) Surgical and non-surgical units as well as intensive and elective care units; (**B**) health facilities located in coal and chemical districts; (**C**) bacteria-contaminated and bacteria-free environments; (**D**) virus-contaminated and virus-free environments. *P* values provided above the bars, Mann–Whitney *U*-test. All samples within each group (*n* = 12 for surgical units; *n* = 16 for non-surgical units; *n* = 10 for intensive care units; *n* = 18 for elective care units; *n* = 11 for coal districts; *n* = 17 for chemical districts; *n* = 20 for bacteria-contaminated samples; *n* = 8 for bacteria-free samples; *n* = 8 for virus-contaminated samples; *n* = 20 for virus-free samples) have been combined in one plot.

**Figure 3 life-11-00639-f003:**
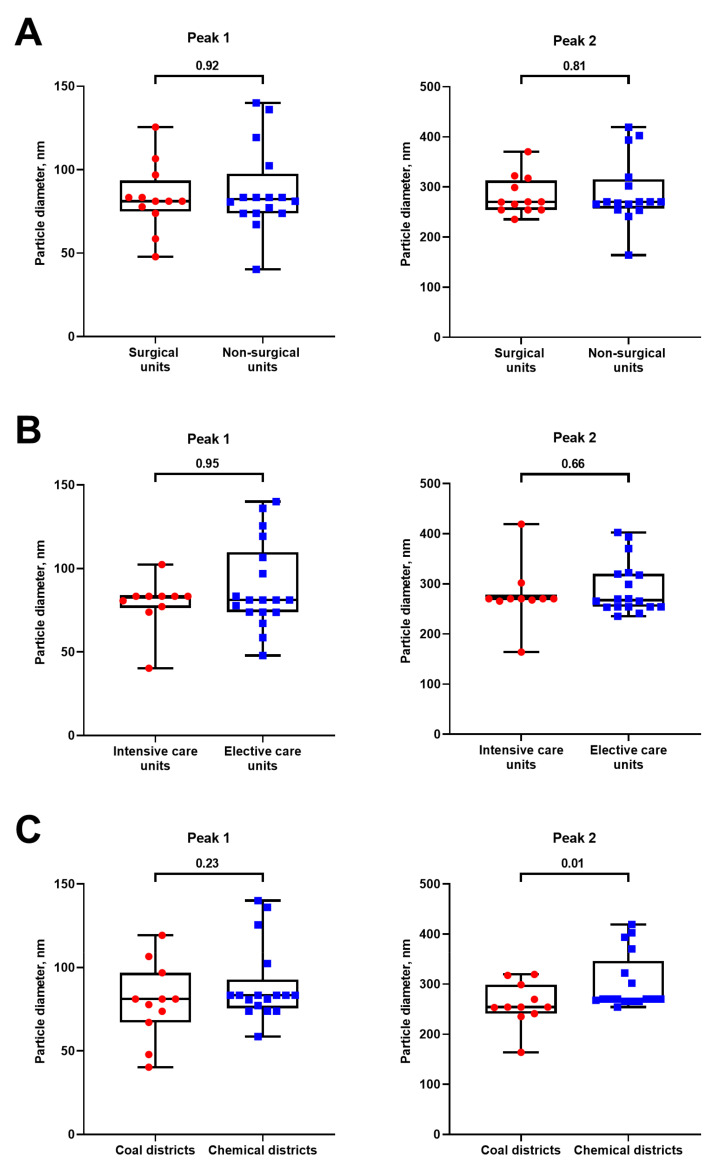
Evaluation of particle-size distribution in Vent-PM samples from the following: (**A**) Surgical and non-surgical units. Red circle dots indicate samples from surgical units while blue square dots indicate those from non-surgical units; (**B**) intensive and elective care units. Red circle dots indicate samples from intensive care units while blue square dots indicate those from elective care units; (**C**) health facilities located in coal and chemical districts. Each dot represents one Vent-PM sample. Red circle dots indicate samples from coal districts while blue square dots indicate those from chemical districts. *P* values provided above the bars, Mann–Whitney *U*-test.

**Figure 4 life-11-00639-f004:**
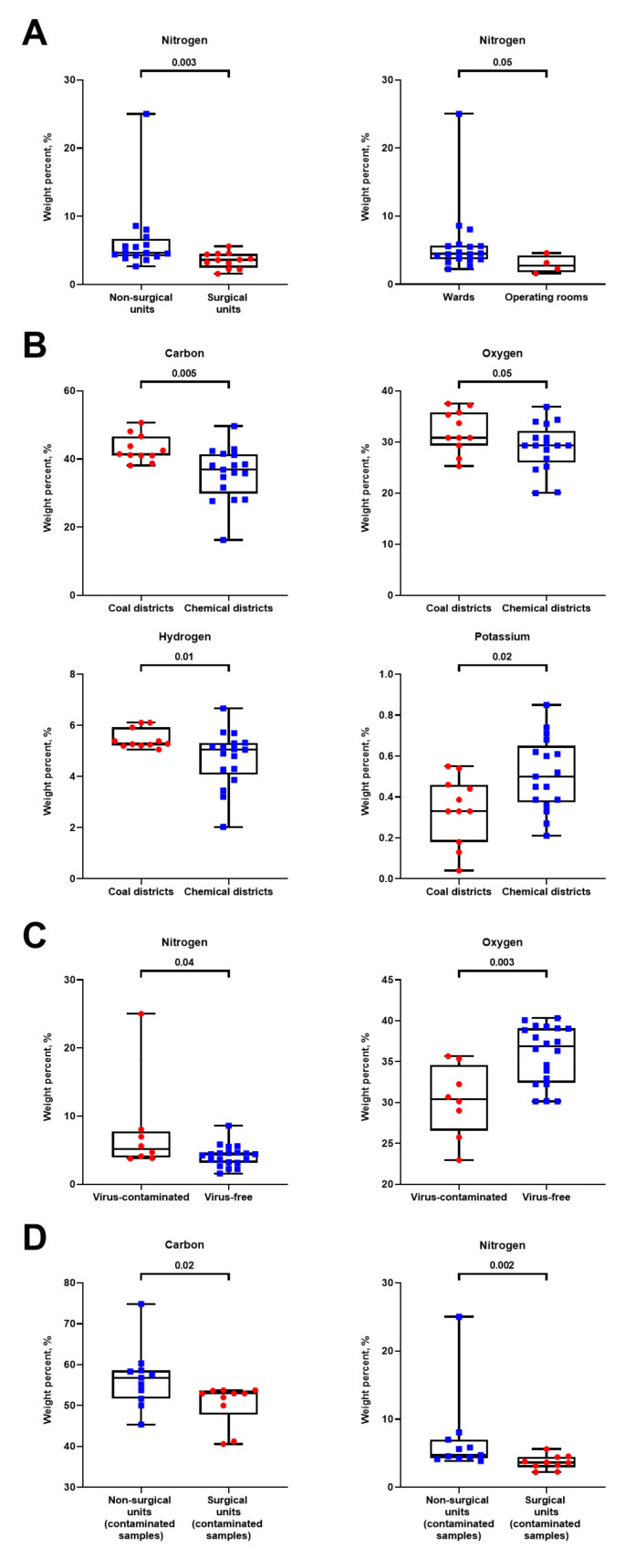
Mineral profiling of the hospital Vent-PM. (**A**) Comparison of nitrogen quantities in non-surgical (blue square dots) vs. surgical (red circle dots) units and wards (blue square dots) vs. operating rooms (red circle dots); (**B**) chemical elements which significantly differed between the health facilities located in coal (red circle dots) and chemical (blue square dots) districts; (**C**) comparison of nitrogen and oxygen in virus-contaminated (red circle dots) and virus-free (blue square dots) PM; (**D**) comparison of carbon and nitrogen in the contaminated Vent-PM from non-surgical (blue square dots) and surgical (red circle dots) units. Each dot represents one Vent-PM sample. *P* values provided above the bars, Mann–Whitney *U*-test.

**Figure 5 life-11-00639-f005:**
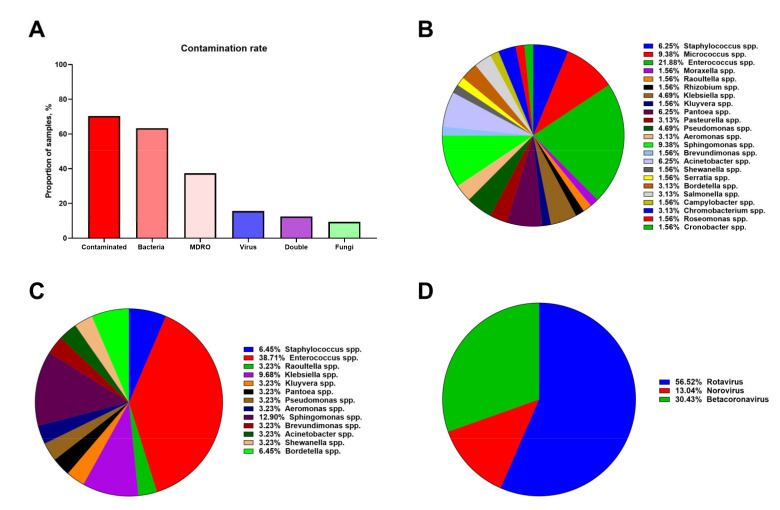
Microbiological profile of hospital Vent-PM (*n* = 128 samples). (**A**) Frequency of Vent-PM contamination by distinct infectious agents. (**B**) Bacterial diversity in Vent-PM. (**C**) Diversity of MDRO in Vent-PM. (**D**) Viral diversity in Vent-PM.

**Figure 6 life-11-00639-f006:**
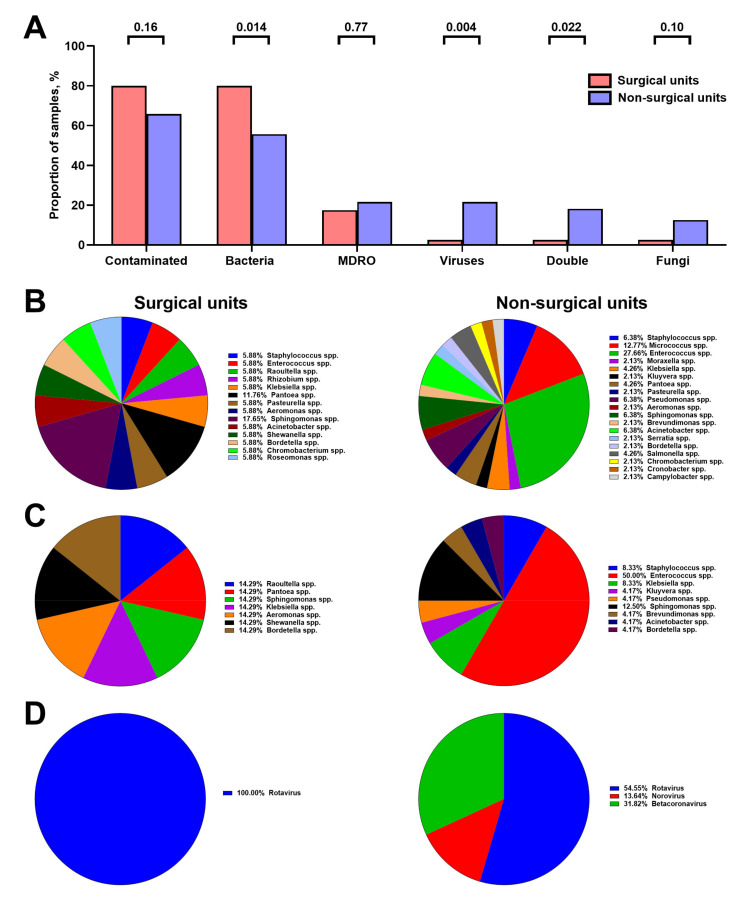
Microbiological profile of hospital Vent-PM from surgical (*n* = 40) and non-surgical (*n* = 88) units. (**A**) Frequency of Vent-PM contamination by distinct infectious agents. *P* values provided above the bars, Mann–Whitney *U*-test. (**B**) Bacterial diversity in Vent-PM. (**C**) Diversity of MDRO in Vent-PM. (**D**) Viral diversity in Vent-PM.

**Figure 7 life-11-00639-f007:**
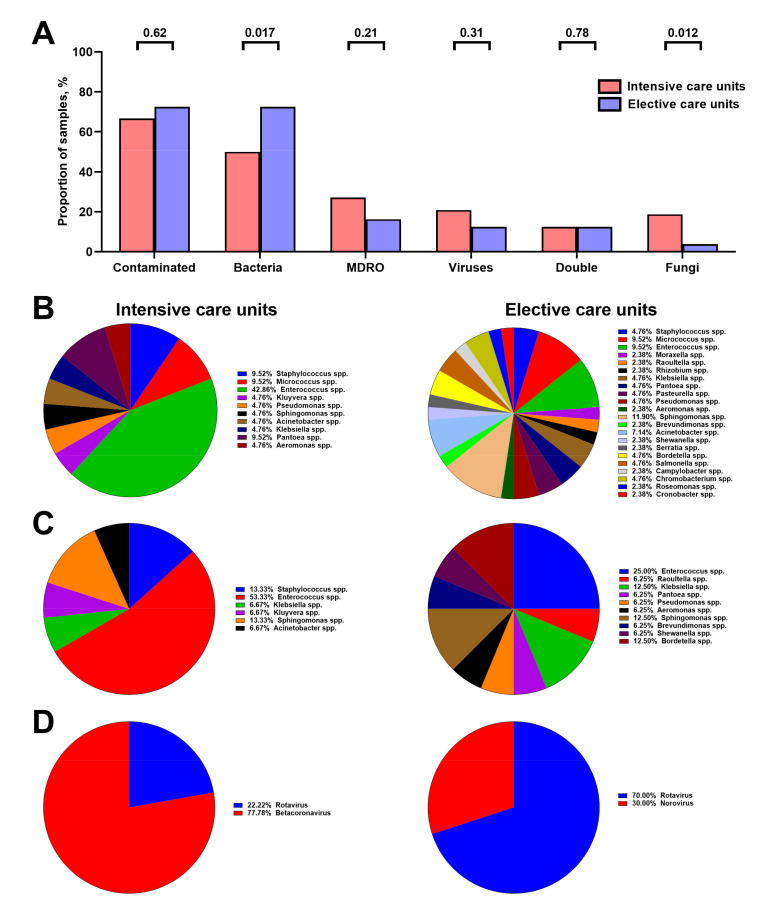
Microbiological profile of hospital Vent-PM from intensive (*n* = 48) and elective (*n* = 80) care units. (**A**) Frequency of Vent-PM contamination by distinct infectious agents. *P* values provided above the bars, Mann–Whitney *U*-test. (**B**) Bacterial diversity in Vent-PM. (**C**) Diversity of MDRO in Vent-PM. (**D**) Viral diversity in Vent-PM.

**Figure 8 life-11-00639-f008:**
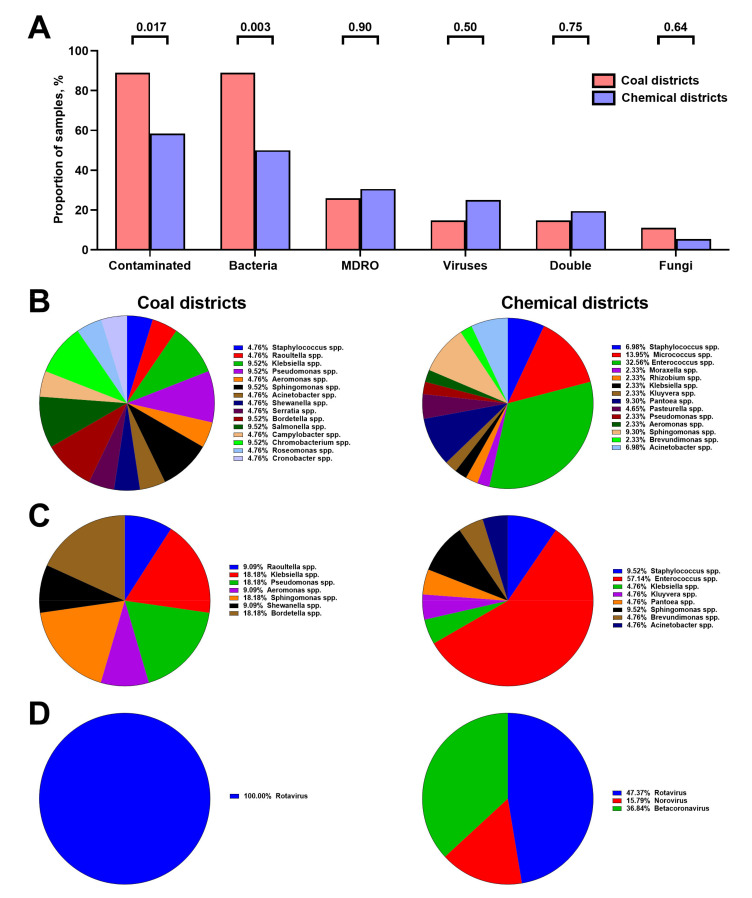
Microbiological profile of hospital Vent-PM from health facilities located in coal (*n* = 27) and chemical (*n* = 36) districts. (**A**) Frequency of Vent-PM contamination by distinct infectious agents. *P* values provided above the bars, Mann–Whitney *U*-test. (**B**) Bacterial diversity in Vent-PM. (**C**) Diversity of MDRO in Vent-PM. (**D**) Viral diversity in Vent-PM.

**Figure 9 life-11-00639-f009:**
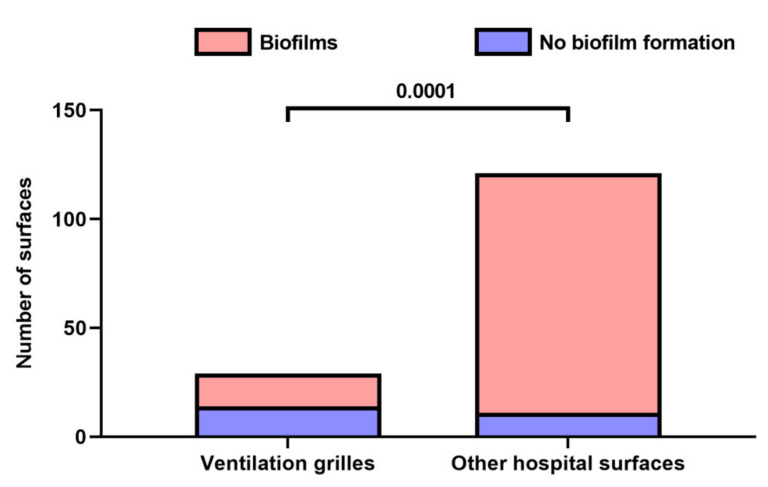
Count of biofilm-positive surfaces at ventilation grilles (*n* = 29) as compared to other hospital surfaces (*n* = 121). *P* value provided above the bars, Mann–Whitney *U*-test.

**Table 1 life-11-00639-t001:** Distribution of Vent-PM across the study groups.

Total (*n* = 128, 13 + 15)						
SU40 (4 + 8)	NSU (88, 9 + 7)	ICU (48, 4 + 6)	ECU (80, 9 + 9)	Coal or chemical districts (63, 13 + 15)		Mixed districts (65)
				Coal 27 (3 + 8)	Chem 36 (10 + 7)	

SU—surgical units; NSU—non-surgical units; ICU—intensive care units; ECU—elective care units. Magenta and orange colours identify the number of samples with globular and fibrillar geometry, respectively.

**Table 2 life-11-00639-t002:** Distribution of Vent-PM samples across the study groups in relation to industrial districts.

				Total (*n* = 128, 13 + 15)							
Coal or Chemical Districts (*n* = 63, 13 + 15)								Mixed Districts (*n* = 65)			
Coal Districts (*n* = 27, 3 + 8)				Chemical Districts (*n* = 36, 10 + 7)							
SU (11, 1 + 5)	NSU (16, 2 + 3)	ICU (3, 0 + 1)	ECU (24, 3 + 7)	SU (6, 3 + 3)	NSU (30, 7 + 4)	ICU (14, 4 + 5)	ECU (22, 6 + 2)	SU (23)	NSU (42)	ICU (31)	ECU (34)

SU—surgical units; NSU—non-surgical units; ICU—intensive care units; ECU—elective care units. Magenta and orange colours identify the number of samples with globular and fibrillar geometry, respectively.

**Table 3 life-11-00639-t003:** Characteristics of the Vent-PM samples.

Sample Number	Collected Area			Sample Size	Sample Collection Date	Investigation Technique
1	NSU	ICU	Chem	≈0.05–0.2 g	15.12.2018	Microbial diversity analysis, SEM
2	SU	ECU	Chem	≈0.05–0.2 g	15.12.2018	Microbial diversity analysis, SEM
3	NSU	ICU	Chem	≈0.05–0.2 g	15.12.2018	Microbial diversity analysis, SEM
4	NSU	ECU	Chem	≈0.05–0.2 g	15.12.2018	Microbial diversity analysis, SEM
5	NSU	ICU	Chem	≈0.05–0.2 g	15.12.2018	Microbial diversity analysis, SEM
6	NSU	ECU	Chem	≈0.05–0.2 g	15.12.2018	Microbial diversity analysis, SEM
7	NSU	ICU	Chem	≈0.05–0.2 g	15.12.2018	Microbial diversity analysis, SEM
8	SU	ECU	Chem	≈0.05–0.2 g	15.12.2018	Microbial diversity analysis, SEM
9	SU	ICU	Chem	≈0.05–0.2 g	15.12.2018	Microbial diversity analysis, SEM
10	SU	ECU	Chem	≈0.05–0.2 g	15.12.2018	Microbial diversity analysis, SEM
11	SU	ECU	Chem	≈0.05–0.2 g	15.12.2018	Microbial diversity analysis, SEM
12	NSU	ECU	Chem	≈0.05–0.2 g	15.12.2018	Microbial diversity analysis, SEM
13	SU	ECU	Coal	≈0.05–0.2 g	25.02.2019	Microbial diversity analysis, SEM
14	SU	ECU	Coal	≈0.05–0.2 g	25.02.2019	Microbial diversity analysis, SEM
15	SU	ECU	Coal	≈0.05–0.2 g	25.02.2019	Microbial diversity analysis, SEM
16	SU	ECU	Coal	≈0.05–0.2 g	25.02.2019	Microbial diversity analysis, SEM
17	SU	ECU	Coal	≈0.05–0.2 g	25.02.2019	Microbial diversity analysis, SEM
18	SU	ECU	Coal	≈0.05–0.2 g	25.02.2019	Microbial diversity analysis, SEM
19	NSU	ICU	Coal	≈0.05–0.2 g	25.02.2019	Microbial diversity analysis
20	NSU	ICU	Coal	≈0.05–0.2 g	25.02.2019	Microbial diversity analysis, SEM
21	NSU	ECU	Coal	≈0.05–0.2 g	25.02.2019	Microbial diversity analysis, SEM
22	NSU	ECU	Coal	≈0.05–0.2 g	25.02.2019	Microbial diversity analysis
23	NSU	ECU	Coal	≈0.05–0.2 g	25.02.2019	Microbial diversity analysis, SEM
24	NSU	ECU	Coal	≈0.05–0.2 g	25.02.2019	Microbial diversity analysis, SEM
25	NSU	ECU	Coal	≈0.05–0.2 g	25.02.2019	Microbial diversity analysis, SEM
26	NSU	ECU	Coal	≈0.05–0.2 g	25.02.2019	Microbial diversity analysis
27	NSU	ICU	Chem	≈0.05–0.2 g	06.03.2019	Microbial diversity analysis, SEM
28	NSU	ICU	Chem	≈0.05–0.2 g	06.03.2019	Microbial diversity analysis, SEM
29	NSU	ICU	Chem	≈0.05–0.2 g	06.03.2019	Microbial diversity analysis, SEM
30	NSU	ICU	Chem	≈0.05–0.2 g	06.03.2019	Microbial diversity analysis, SEM
31	SU	ECU	Chem	≈0.05–0.2 g	06.03.2019	Microbial diversity analysis, SEM
32	NSU	ECU	Chem	≈0.05–0.2 g	12.03.2019	Microbial diversity analysis
33	NSU	ECU	Chem	≈0.05–0.2 g	12.03.2019	Microbial diversity analysis
34	NSU	ICU	Chem	≈0.05–0.2 g	12.03.2019	Microbial diversity analysis
35	NSU	ICU	Chem	≈0.05–0.2 g	12.03.2019	Microbial diversity analysis
36	NSU	ECU	Chem	≈0.05–0.2 g	12.03.2019	Microbial diversity analysis
37	NSU	ECU	Chem	≈0.05–0.2 g	12.03.2019	Microbial diversity analysis
38	NSU	ECU	Chem	≈0.05–0.2 g	12.03.2019	Microbial diversity analysis
39	NSU	ECU	Chem	≈0.05–0.2 g	12.03.2019	Microbial diversity analysis
40	NSU	ECU	Chem	≈0.05–0.2 g	12.03.2019	Microbial diversity analysis
41	NSU	ECU	Chem	≈0.05–0.2 g	12.03.2019	Microbial diversity analysis
42	NSU	ECU	Coal	≈0.05–0.2 g	11.09.2019	Microbial diversity analysis
43	NSU	ECU	Coal	≈0.05–0.2 g	11.09.2019	Microbial diversity analysis
44	NSU	ECU	Coal	≈0.05–0.2 g	11.09.2019	Microbial diversity analysis
45	NSU	ICU	Coal	≈0.05–0.2 g	11.09.2019	Microbial diversity analysis
46	NSU	ECU	Chem	≈0.05–0.2 g	25.02.2020	Microbial diversity analysis
47	NSU	ECU	Chem	≈0.05–0.2 g	25.02.2020	Microbial diversity analysis
48	NSU	ECU	Chem	≈0.05–0.2 g	25.02.2020	Microbial diversity analysis
49	NSU	ICU	Chem	≈0.05–0.2 g	25.02.2020	Microbial diversity analysis
50	NSU	ICU	Chem	≈0.05–0.2 g	25.02.2020	Microbial diversity analysis
51	NSU	ICU	Chem	≈0.05–0.2 g	25.02.2020	Microbial diversity analysis
52	NSU	ECU	Chem	≈0.05–0.2 g	25.02.2020	Microbial diversity analysis
53	NSU	ECU	Chem	≈0.05–0.2 g	25.02.2020	Microbial diversity analysis
54	NSU	ECU	Chem	≈0.05–0.2 g	25.02.2020	Microbial diversity analysis
55	SU	ECU	Coal	≈0.05–0.2 g	11.09.2019	Microbial diversity analysis
56	SU	ECU	Coal	≈0.05–0.2 g	11.09.2019	Microbial diversity analysis
57	SU	ECU	Coal	≈0.05–0.2 g	11.09.2019	Microbial diversity analysis
58	SU	ECU	Coal	≈0.05–0.2 g	11.09.2019	Microbial diversity analysis
59	SU	ECU	Coal	≈0.05–0.2 g	11.09.2019	Microbial diversity analysis
60	NSU	ECU	Coal	≈0.05–0.2 g	11.09.2019	Microbial diversity analysis
61	NSU	ECU	Coal	≈0.05–0.2 g	11.09.2019	Microbial diversity analysis
62	NSU	ECU	Coal	≈0.05–0.2 g	11.09.2019	Microbial diversity analysis
63	NSU	ECU	Coal	≈0.05–0.2 g	11.09.2019	Microbial diversity analysis
64	NSU	ECU	Mixed	≈0.05–0.2 g	16.08.2020	Microbial diversity analysis
65	NSU	ECU	Mixed	≈0.05–0.2 g	16.08.2020	Microbial diversity analysis
66	NSU	ECU	Mixed	≈0.05–0.2 g	16.08.2020	Microbial diversity analysis
67	NSU	ECU	Mixed	≈0.05–0.2 g	16.08.2020	Microbial diversity analysis
68	NSU	ECU	Mixed	≈0.05–0.2 g	16.08.2020	Microbial diversity analysis
69	NSU	ECU	Mixed	≈0.05–0.2 g	16.08.2020	Microbial diversity analysis
70	NSU	ECU	Mixed	≈0.05–0.2 g	16.08.2020	Microbial diversity analysis
71	NSU	ECU	Mixed	≈0.05–0.2 g	16.08.2020	Microbial diversity analysis
72	SU	ECU	Mixed	≈0.05–0.2 g	16.08.2020	Microbial diversity analysis
73	NSU	ECU	Mixed	≈0.05–0.2 g	16.08.2020	Microbial diversity analysis
74	NSU	ECU	Mixed	≈0.05–0.2 g	16.08.2020	Microbial diversity analysis
75	NSU	ECU	Mixed	≈0.05–0.2 g	21.09.2020	Microbial diversity analysis
76	SU	ECU	Mixed	≈0.05–0.2 g	21.09.2020	Microbial diversity analysis
77	SU	ECU	Mixed	≈0.05–0.2 g	21.09.2020	Microbial diversity analysis
78	SU	ECU	Mixed	≈0.05–0.2 g	21.09.2020	Microbial diversity analysis
79	SU	ECU	Mixed	≈0.05–0.2 g	21.09.2020	Microbial diversity analysis
80	SU	ECU	Mixed	≈0.05–0.2 g	21.09.2020	Microbial diversity analysis
81	SU	ECU	Mixed	≈0.05–0.2 g	21.09.2020	Microbial diversity analysis
82	SU	ECU	Mixed	≈0.05–0.2 g	21.09.2020	Microbial diversity analysis
83	SU	ECU	Mixed	≈0.05–0.2 g	21.09.2020	Microbial diversity analysis
84	SU	ECU	Mixed	≈0.05–0.2 g	21.09.2020	Microbial diversity analysis
85	SU	ECU	Mixed	≈0.05–0.2 g	21.09.2020	Microbial diversity analysis
86	SU	ECU	Mixed	≈0.05–0.2 g	21.09.2020	Microbial diversity analysis
87	SU	ECU	Mixed	≈0.05–0.2 g	21.09.2020	Microbial diversity analysis
88	SU	ECU	Mixed	≈0.05–0.2 g	21.09.2020	Microbial diversity analysis
89	SU	ECU	Mixed	≈0.05–0.2 g	21.09.2020	Microbial diversity analysis
90	SU	ECU	Mixed	≈0.05–0.2 g	21.09.2020	Microbial diversity analysis
91	SU	ECU	Mixed	≈0.05–0.2 g	21.09.2020	Microbial diversity analysis
92	SU	ECU	Mixed	≈0.05–0.2 g	21.09.2020	Microbial diversity analysis
93	SU	ECU	Mixed	≈0.05–0.2 g	21.09.2020	Microbial diversity analysis
94	SU	ECU	Mixed	≈0.05–0.2 g	21.09.2020	Microbial diversity analysis
95	SU	ECU	Mixed	≈0.05–0.2 g	21.09.2020	Microbial diversity analysis
96	SU	ECU	Mixed	≈0.05–0.2 g	21.09.2020	Microbial diversity analysis
97	SU	ECU	Mixed	≈0.05–0.2 g	01.04.2021	Microbial diversity analysis
98	NSU	ICU	Mixed	≈0.05–0.2 g	01.04.2021	Microbial diversity analysis
99	NSU	ICU	Mixed	≈0.05–0.2 g	01.04.2021	Microbial diversity analysis
100	NSU	ICU	Mixed	≈0.05–0.2 g	01.04.2021	Microbial diversity analysis
101	NSU	ICU	Mixed	≈0.05–0.2 g	01.04.2021	Microbial diversity analysis
102	NSU	ICU	Mixed	≈0.05–0.2 g	01.04.2021	Microbial diversity analysis
103	NSU	ICU	Mixed	≈0.05–0.2 g	01.04.2021	Microbial diversity analysis
104	NSU	ICU	Mixed	≈0.05–0.2 g	01.04.2021	Microbial diversity analysis
105	NSU	ICU	Mixed	≈0.05–0.2 g	01.04.2021	Microbial diversity analysis
106	NSU	ICU	Mixed	≈0.05–0.2 g	01.04.2021	Microbial diversity analysis
107	NSU	ICU	Mixed	≈0.05–0.2 g	01.04.2021	Microbial diversity analysis
108	NSU	ICU	Mixed	≈0.05–0.2 g	01.04.2021	Microbial diversity analysis
109	NSU	ICU	Mixed	≈0.05–0.2 g	01.04.2021	Microbial diversity analysis
110	NSU	ICU	Mixed	≈0.05–0.2 g	01.04.2021	Microbial diversity analysis
111	NSU	ICU	Mixed	≈0.05–0.2 g	01.04.2021	Microbial diversity analysis
112	NSU	ICU	Mixed	≈0.05–0.2 g	01.04.2021	Microbial diversity analysis
113	NSU	ICU	Mixed	≈0.05–0.2 g	01.04.2021	Microbial diversity analysis
114	NSU	ICU	Mixed	≈0.05–0.2 g	01.04.2021	Microbial diversity analysis
115	NSU	ICU	Mixed	≈0.05–0.2 g	01.04.2021	Microbial diversity analysis
116	NSU	ICU	Mixed	≈0.05–0.2 g	01.04.2021	Microbial diversity analysis
117	NSU	ICU	Mixed	≈0.05–0.2 g	01.04.2021	Microbial diversity analysis
118	NSU	ICU	Mixed	≈0.05–0.2 g	01.04.2021	Microbial diversity analysis
119	NSU	ICU	Mixed	≈0.05–0.2 g	01.04.2021	Microbial diversity analysis
120	NSU	ICU	Mixed	≈0.05–0.2 g	01.04.2021	Microbial diversity analysis
121	NSU	ICU	Mixed	≈0.05–0.2 g	01.04.2021	Microbial diversity analysis
122	NSU	ICU	Mixed	≈0.05–0.2 g	01.04.2021	Microbial diversity analysis
123	NSU	ICU	Mixed	≈0.05–0.2 g	01.04.2021	Microbial diversity analysis
124	NSU	ICU	Mixed	≈0.05–0.2 g	01.04.2021	Microbial diversity analysis
125	NSU	ICU	Mixed	≈0.05–0.2 g	01.04.2021	Microbial diversity analysis
126	NSU	ICU	Mixed	≈0.05–0.2 g	01.04.2021	Microbial diversity analysis
127	NSU	ICU	Mixed	≈0.05–0.2 g	01.04.2021	Microbial diversity analysis
128	NSU	ICU	Mixed	≈0.05–0.2 g	01.04.2021	Microbial diversity analysis

SU—surgical units; NSU—non-surgical units; ICU—intensive care units; ECU—elective care units.

**Table 4 life-11-00639-t004:** Contamination of the Vent-PM samples in relation to the study groups.

Sample Number	Collected Area			Contamination Status	Bacteria	MDRO	Viruses	Fungi
1	NSU	ICU	Chem	+	+	+	+	-
2	SU	ECU	Chem	-	-	-	-	-
3	NSU	ICU	Chem	+	+	+	-	-
4	NSU	ECU	Chem	-	-	-	-	-
5	NSU	ICU	Chem	-	-	-	-	-
6	NSU	ECU	Chem	-	-	-	-	-
7	NSU	ICU	Chem	+	-	-	+	-
8	SU	ECU	Chem	+	+	-	-	-
9	SU	ICU	Chem	+	+	+	-	-
10	SU	ECU	Chem	+	+	+	-	-
11	SU	ECU	Chem	+	+	-	-	-
12	NSU	ECU	Chem	+	+	-	+	-
13	SU	ECU	Coal	+	+	+	-	-
14	SU	ECU	Coal	+	+	+	+	-
15	SU	ECU	Coal	+	+	-	-	-
16	SU	ECU	Coal	+	+	+	-	-
17	SU	ECU	Coal	+	+	+	-	-
18	SU	ECU	Coal	+	+	-	-	-
19	NSU	ICU	Coal	-	-	-	-	-
20	NSU	ICU	Coal	+	+	-	-	-
21	NSU	ECU	Coal	+	+	+	-	-
22	NSU	ECU	Coal	-	-	-	-	-
23	NSU	ECU	Coal	+	+	+	+	-
24	NSU	ECU	Coal	+	+	-	+	-
25	NSU	ECU	Coal	+	+	-	+	+
26	NSU	ECU	Coal	+	+	-	-	-
27	NSU	ICU	Chem	+	-	-	-	+
28	NSU	ICU	Chem	-	-	-	-	-
29	NSU	ICU	Chem	-	-	-	-	-
30	NSU	ICU	Chem	+	-	-	+	+
31	SU	ECU	Chem	-	-	-	-	-
32	NSU	ECU	Chem	-	-	-	-	-
33	NSU	ECU	Chem	+	+	-	-	-
34	NSU	ICU	Chem	+	+	-	-	-
35	NSU	ICU	Chem	-	-	-	-	-
36	NSU	ECU	Chem	-	-	-	-	-
37	NSU	ECU	Chem	-	-	-	-	-
38	NSU	ECU	Chem	-	-	-	-	-
39	NSU	ECU	Chem	-	-	-	-	-
40	NSU	ECU	Chem	-	-	-	-	-
41	NSU	ECU	Chem	-	-	-	-	-
42	NSU	ECU	Coal	+	+	-	-	-
43	NSU	ECU	Coal	+	+	+	-	+
44	NSU	ECU	Coal	+	+	-	-	-
45	NSU	ICU	Coal	-	-	-	-	-
46	NSU	ECU	Chem	+	+	-	+	-
47	NSU	ECU	Chem	+	+	+	+	-
48	NSU	ECU	Chem	+	+	+	+	-
49	NSU	ICU	Chem	+	+	+	-	-
50	NSU	ICU	Chem	+	+	+	-	-
51	NSU	ICU	Chem	+	+	+	-	-
52	NSU	ECU	Chem	+	+	+	+	-
53	NSU	ECU	Chem	+	+	+	+	-
54	NSU	ECU	Chem	+	+	-	-	-
55	SU	ECU	Coal	+	+	-	-	-
56	SU	ECU	Coal	+	+	-	-	-
57	SU	ECU	Coal	+	+	-	-	+
58	SU	ECU	Coal	+	+	-	-	-
59	SU	ECU	Coal	+	+	-	-	-
60	NSU	ECU	Coal	+	+	-	-	-
61	NSU	ECU	Coal	+	+	-	-	-
62	NSU	ECU	Coal	+	+	-	-	-
63	NSU	ECU	Coal	+	+	-	-	-
64	NSU	ECU	Mixed	-	-	-	-	-
65	NSU	ECU	Mixed	-	-	-	-	-
66	NSU	ECU	Mixed	-	-	-	-	-
67	NSU	ECU	Mixed	+	+	-	-	-
68	NSU	ECU	Mixed	+	+	-	-	-
69	NSU	ECU	Mixed	+	+	-	-	-
70	NSU	ECU	Mixed	+	+	-	-	-
71	NSU	ECU	Mixed	+	+	-	-	-
72	SU	ECU	Mixed	+	+	-	-	-
73	NSU	ECU	Mixed	+	+	-	-	-
74	NSU	ECU	Mixed	-	-	-	-	-
75	NSU	ECU	Mixed	+	+	-	-	-
76	SU	ECU	Mixed	-	-	-	-	-
77	SU	ECU	Mixed	+	+	-	-	-
78	SU	ECU	Mixed	+	+	-	-	-
79	SU	ECU	Mixed	+	+	-	-	-
80	SU	ECU	Mixed	+	+	-	-	-
81	SU	ECU	Mixed	+	+	-	-	-
82	SU	ECU	Mixed	+	+	+	-	-
83	SU	ECU	Mixed	+	+	-	-	-
84	SU	ECU	Mixed	+	+	-	-	-
85	SU	ECU	Mixed	-	-	-	-	-
86	SU	ECU	Mixed	+	+	-	-	-
87	SU	ECU	Mixed	+	+	-	-	-
88	SU	ECU	Mixed	+	+	-	-	-
89	SU	ECU	Mixed	+	+	-	-	-
90	SU	ECU	Mixed	+	+	-	-	-
91	SU	ECU	Mixed	+	+	-	-	-
92	SU	ECU	Mixed	-	-	-	-	-
93	SU	ECU	Mixed	-	-	-	-	-
94	SU	ECU	Mixed	+	+	-	-	-
95	SU	ECU	Mixed	+	+	-	-	-
96	SU	ECU	Mixed	-	-	-	-	-
97	SU	ECU	Mixed	-	-	-	-	-
98	NSU	ICU	Mixed	+	+	+	+	-
99	NSU	ICU	Mixed	-	-	-	-	-
100	NSU	ICU	Mixed	+	+	+	-	-
101	NSU	ICU	Mixed	+	-	-	-	+
102	NSU	ICU	Mixed	-	-	-	-	-
103	NSU	ICU	Mixed	-	-	-	-	-
104	NSU	ICU	Mixed	+	+	+	+	+
105	NSU	ICU	Mixed	+	+	+	-	-
106	NSU	ICU	Mixed	+	-	-	+	+
107	NSU	ICU	Mixed	-	-	-	-	-
108	NSU	ICU	Mixed	-	-	-	-	-
109	NSU	ICU	Mixed	+	+	+	-	-
110	NSU	ICU	Mixed	+	+	+	+	-
111	NSU	ICU	Mixed	+	+	-	+	-
112	NSU	ICU	Mixed	+	+	+	-	-
113	NSU	ICU	Mixed	+	+	-	+	-
114	NSU	ICU	Mixed	+	+	-	+	-
115	NSU	ICU	Mixed	+	+	-	-	-
116	NSU	ICU	Mixed	-	-	-	-	-
117	NSU	ICU	Mixed	-	-	-	-	-
118	NSU	ICU	Mixed	+	+	-	-	-
119	NSU	ICU	Mixed	+	+	-	-	+
120	NSU	ICU	Mixed	+	+	-	-	-
121	NSU	ICU	Mixed	-	-	-	-	-
122	NSU	ICU	Mixed	+	+	-	-	-
123	NSU	ICU	Mixed	+	-	-	-	+
124	NSU	ICU	Mixed	-	-	-	-	-
125	NSU	ICU	Mixed	+	-	-	-	+
126	NSU	ICU	Mixed	-	-	-	-	-
127	NSU	ICU	Mixed	+	-	-	-	+
128	NSU	ICU	Mixed	+	+	-	-	-

SU—surgical units; NSU—non-surgical units; ICU—intensive care units; ECU—elective care units; MDRO—multidrug-resistant organisms.

## Data Availability

The datasets generated and/or analysed during the current study are available from the corresponding author upon reasonable request.

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
