# Peer review of "Ventilation-Associated Particulate Matter Is a Potential Reservoir of Multidrug-Resistant Organisms in Health Facilities"

_life, 2021, doi:10.3390/life11070639_

Round 1
Reviewer 1 Report
- The study “Ventilation – associated particulate matter is a potential reservoir of multidrug-resistant organisms in health facilities” provide a very important contribution to understand how Vent-PM can be a reservoir of healthcare-associated air-borne infectious by multidrug-resistant organisms in the hospital environment
- Multidrug-resistant pathogens represent a substantial proportion of nosocomial infections in healthcare facilities
- In recent years, bacteria, fungi and virus have increased the relevance as multidrug-resistant microorganisms for nosocomial infections in immunocompetent patients as well as for elderly adults and those with chronic medical conditions which numbers are increasing
- In recent years, infection control in healthcare facilities has seen many advances, including rapid molecular screening tests for resistant microorganisms
- In this study the authors provided information for molecular tests for identification of virus and bacteria, and cultural methods for the identification of bacterial diversity. However there is no information about the specific methodology (molecular and cultural) for fungi
- Aspergillus spp have increased relevance in recent years as multidrug-resistant pathogen with a great risk for nosocomial infections in those immunocompetent patients as well as in those immunocompromised and with other risk factors (chronic obstructive pulmonary disease, liver cirrhosis, and diabetes), including the patients who underwent abdominal surgical interventions and those on peritoneal dialysis
- So, I wonder to understand why the authors do not identified Aspergillus spp in this study, and which molecular and cultural methods they have used for fungi identification
Author Response
Q1: Please summarize the main findings of the study. The study “Ventilation – associated particulate matter is a potential reservoir of multidrug-resistant organisms in health facilities” provide a very important contribution to understand how Vent-PM can be a reservoir of healthcare-associated air-borne infectious by multidrug-resistant organisms in the hospital environment. Multidrug-resistant pathogens represent a substantial proportion of nosocomial infections in healthcare facilities. In recent years, bacteria, fungi and virus have increased the relevance as multidrug-resistant microorganisms for nosocomial infections in immunocompetent patients as well as for elderly adults and those with chronic medical conditions which numbers are increasing. In recent years, infection control in healthcare facilities has seen many advances, including rapid molecular screening tests for resistant microorganisms.
A1: We sincerely thank the reviewer for the helpful comments and constructive criticism. The respective comments are provided below and the manuscript has been correspondingly revised in order to address all points indicated by the reviewer.
___________________________________________________________________________
Q2: In this study the authors provided information for molecular tests for identification of virus and bacteria, and cultural methods for the identification of bacterial diversity. However there is no information about the specific methodology (molecular and cultural) for fungi. Aspergillus spp have increased relevance in recent years as multidrug-resistant pathogen with a great risk for nosocomial infections in those immunocompetent patients as well as in those immunocompromised and with other risk factors (chronic obstructive pulmonary disease, liver cirrhosis, and diabetes), including the patients who underwent abdominal surgical interventions and those on peritoneal dialysis. So, I wonder to understand why the authors do not identified Aspergillus spp in this study, and which molecular and cultural methods they have used for fungi identification.
A2: We agree with the reviewer that the techniques of fungal identification have not been sufficiently described in the manuscript. To detect pathogenic yeasts and molds, Vent-PM samples were resuspended in Sabouraud dextrose broth (M033, HiMedia Laboratories, Mumbai, India), incubated for 24 hours at 35°C, then inoculated into Sabouraud dextrose agar (M063, HiMedia Laboratories, Mumbai, India) and CHROMagar Candida (CA223-25, CHROMagar, Paris, France) and finally incubated for 7 days at 35°C. Identification of pathogenic yeasts was carried out by VITEK® 2 YST identification cards (21343, BioMerieux, Marcy-l'Etoile, France) employing VITEK® 2 COMPACT Microbial Detection System (BioMerieux, Marcy-l'Etoile, France). Molds were detected by a microscopic examination. We added this information into the respective section of Materials and Methods (2.4 Fungal evaluation).
As pathogenic yeasts (including Candida spp.) have not been detected, fungal diversity was limited to molds. Unexpectedly, microscopic examination also did not reveal Aspergillus spp. in Vent-PM samples. Possible explanations include extremely cold winter in Siberia (-45°C in January) in combination with low air humidity (≈ 30%) in healthcare facilities due to the central heating system working ≈ 8 months annually. We added this information into Results and Discussion.
Reviewer 2 Report
In this work, the Authors performed a systematic study for the evaluation of vent-PM as vector/reservoir of Viruses and Bacterial pathogens (including antibiotic resistant and biofilm forming strains). The Authors also tried to associate the diversity of contaminants with the origin of PM (Surgical/not Surgical units, Intensive/Elective care units, Coal/Chemical districts). The work in context appears very interesting, although I suggest a reorganization in some sections, particularly those concerning sampling.
Major considerations
- The work presents a considerable number of dates, for which I have found very difficult to understand sampling. In this context, the description of sampling should be reorganized for a better view of the results by the reader. It is not clear:
- The number of total samples analysed;
- If I understand correctly, some samples may belong to two different categories, for example to both surgical and intensive care units. Therefore, a solution could be to added a table after line 102 (see Attachment)
- Material and Methods section should be divided in subsection with related title. For example: paragraphs from line 97 to line 122 could be “Sampling and PM chemical evaluation”; paragraphs from line 123 to 131 “Viruses evaluation”; from 132 to 151 “Bacterial evaluation”; from 151 to 156 “Statistical Analysis”
- Line 322-327 and 332-334 are Consideration/Discussion of the presence of biofilm in Vent-PM, therefore I suggest to move them in the Discussion Section.
- Supplementary Figure 2 should be moved in the manuscript for a better view of the results about biofilm by the reader.
- Similarly to MDRO, was the detected biofilms associated with specific bacterial strains.This is important because in this case for biofilm-producing strains PM could represent a long-term reservoir compared to other pathogens.
Minor considerations
- Figure resolution should be improved.
- In line 49, I suggest to change “antimicrobials” with “antibiotics”. In fact, although all antibiotics are antimicrobials; not all antimicrobials are antibiotics and multidrug-resistance is generally referred to antibiotic resistance.
- When they first appear, PM should be specified as size range: PM2.5-10 (particulate matter in the size ranges 2.5-10 micro m), etc. This is needed for readers with little familiarity about PM labelling.

Author Response
Q1: Please summarize the main findings of the study. In this work, the Authors performed a systematic study for the evaluation of vent-PM as vector/reservoir of Viruses and Bacterial pathogens (including antibiotic resistant and biofilm forming strains). The Authors also tried to associate the diversity of contaminants with the origin of PM (Surgical/not Surgical units, Intensive/Elective care units, Coal/Chemical districts). The work in context appears very interesting, although I suggest a reorganization in some sections, particularly those concerning sampling.
A1: We sincerely thank the reviewer for the helpful comments and constructive criticism. The respective comments are provided below and the manuscript has been correspondingly revised in order to address all points indicated by the reviewer.
___________________________________________________________________________
Q2: The work presents a considerable number of dates, for which I have found very difficult to understand sampling. In this context, the description of sampling should be reorganized for a better view of the results by the reader. It is not clear:
- The number of total samples analysed;
- If I understand correctly, some samples may belong to two different categories, for example to both surgical and intensive care units. Therefore, a solution could be to added a table after line 102 (see Attachment).
A2: We agree with the reviewer that the description of sampling was not sufficiently clear for the readers. Indeed, distinct PM samples could belong to both surgical and intensive care units. Regarding the ecological impact, 63 samples were collected in healthcare facilities located within the districts having exclusively coal mines or chemical plants, while 65 samples were gathered in the territories with both types of industrial enterprises (mixed districts). To address the hypothesis that the microbial composition of PM from the hospital ventilation systems (Vent-PM) depends on the features of outdoor air pollution, we specifically compared Vent-PM samples from coal and chemical districts. We also agree that the manuscript must contain a table indicating the sample distribution across the study groups. Yet, we think that sample distribution is better for understanding if presented in two different tables, where the first one illustrates the total distribution and the second shows the distribution in relation to industrial districts. We therefore included such tables in the revised manuscript (Revised Table 1 and Revised Table 2), strictly adhering to the suggestion of the reviewer.
_____________________________________________________________________________
Q3: Material and Methods section should be divided in subsection with related title. For example: paragraphs from line 97 to line 122 could be “Sampling and PM chemical evaluation”; paragraphs from line 123 to 131 “Viruses evaluation”; from 132 to 151 “Bacterial evaluation”; from 151 to 156 “Statistical Analysis”
A3: We agree with the reviewer that the subsectioning contributes to clarification of the Materials and Methods section and divided it into subsections according to the reviewer’s suggestion.
_____________________________________________________________________________
Q4: Line 322-327 and 332-334 are Consideration/Discussion of the presence of biofilm in Vent-PM, therefore I suggest to move them in the Discussion Section. Supplementary Figure 2 should be moved in the manuscript for a better view of the results about biofilm by the reader.
A4: We agree with the reviewer with regards to these issues and moved the indicated text into the Discussion. We have also moved Supplementary Figure 2 into the main text (Revised Figure 9).
_____________________________________________________________________________
Q5: Similarly to MDRO, was the detected biofilms associated with specific bacterial strains. This is important because in this case for biofilm-producing strains PM could represent a long-term reservoir compared to other pathogens.
A5: We agree with the reviewer that specific multidrug-resistant strains are particularly important in context of augmented biofilm formation. Indeed, Vent-PM from 6 out of 14 biofilms revealed on 29 ventilation grilles harboured multidrug-resistant bacterial strains, namely Roseomonas gelardii, Serratia plymutica, Sphingomonas pacimobilis, Enterococcus faecium, and Klebsiella pneumoniae (2). We indicated this in the respective paragraph of the Results.
_____________________________________________________________________________
Q6: Minor considerations 1) Figure resolution should be improved. 2) In line 49, I suggest to change “antimicrobials” with “antibiotics”. In fact, although all antibiotics are antimicrobials; not all antimicrobials are antibiotics and multidrug-resistance is generally referred to antibiotic resistance. 3) When they first appear, PM should be specified as size range: PM2.5-10 (particulate matter in the size ranges 2.5-10 micro m), etc. This is needed for readers with little familiarity about PM labelling.
A6: We agree with the reviewer with all considerations indicated.
1) We are a bit puzzled about problems with figure resolution as all figures have been saved directly from the latest version of GraphPad Prism or electron microscope. Probably, the resolution has been lowered during the PDF conversion and we will control this issue if our paper is accepted for publication.
2) We replaced “antimicrobials” with “antibiotics” in all cases where we assessed or discussed multidrug resistance.
3) We added the size specifications for each type of PM indicated in the Introduction.
Reviewer 3 Report
The authors investigated the ventilation-associated particulate matter as a potential reservoir of multidrug-resistant organisms in health facilities. The author's employed EM and microbial experiments, however, no details on metadata fail to address the problem statement that PM of ventilation can carry MDRO.
Major comments
Detailed sample collection and distribution for various analyses were missing. Without the proper metadata on samples, it is hard to understand the results provided in this paper.
From paper: For visualisation, Vent-PM samples (n = 16 for non-surgical and n = 12 for surgical units, n = 12 for elective care and n = 16 for intensive care units, n = 11 for coal and n = 17 for chemical districts). However, I am unable to understand when the sample collected? 16 sample for non-surgical means, the sample from sample area, same day or different days? How they processed for various analysis details?
So, I would suggest the author provide complete information of samples in a tabular format (Sample number, collected area, sample size [collected amount – for eg., swab – 25 cm2, air sample 1 m3 and/or vent filter area], sample collection date, provide which sample carried for SEM and Microbial diversity analysis with sample ID and numbers).
Figure 1: The authors showed only one or two EM images and it has been shown in figure 2. However, details are missing in Figure 1 and 2. For SEM, provide elementary information with the magnification and voltage used. For figure 2 how authors combined all samples (eg., n=16 samples in one bar plot).
Figure 3 and 4: box-plot has points but no details which points belongs to which samples more confusing.
Figure 5: All samples were pooled? Authors should provide how individual sample has microbial diversity and distributions.
From the paper: Vent-PM from the surgical units were most frequently contaminated by bacteria 242 (32/40 samples, 80.0%), with only a minor proportion of MDRO (7/40, 17.5% - which 7 samples from the original sample?? So provide separate table do this sample has lots of PM contaminants or not???) and rare 243 (1/40, 2.5%) contamination by viruses or fungi (Figure 6A)
Author Response
Q1: Please summarize the main findings of the study. The authors investigated the ventilation-associated particulate matter as a potential reservoir of multidrug-resistant organisms in health facilities. The author's employed EM and microbial experiments, however, no details on metadata fail to address the problem statement that PM of ventilation can carry MDRO.
A1: We sincerely thank the reviewer for the helpful comments and constructive criticism. The respective comments are provided below and the manuscript has been correspondingly revised in order to address all points indicated by the reviewer.
___________________________________________________________________________
Q2: Detailed sample collection and distribution for various analyses were missing. Without the proper metadata on samples, it is hard to understand the results provided in this paper.
From paper: For visualisation, Vent-PM samples (n = 16 for non-surgical and n = 12 for surgical units, n = 12 for elective care and n = 16 for intensive care units, n = 11 for coal and n = 17 for chemical districts). However, I am unable to understand when the sample collected? 16 sample for non-surgical means, the sample from sample area, same day or different days? How they processed for various analysis details?
So, I would suggest the author provide complete information of samples in a tabular format (Sample number, collected area, sample size [collected amount – for eg., swab – 25 cm2, air sample 1 m3 and/or vent filter area], sample collection date, provide which sample carried for SEM and Microbial diversity analysis with sample ID and numbers).
A2: We agree with the reviewer that the metadata have not been well presented. In the revised manuscript, we provide a table (Revised Table 3) in strict adherence to the reviewer’s suggestion.
_____________________________________________________________________________
Q3: Figure 1: The authors showed only one or two EM images and it has been shown in figure 2. However, details are missing in Figure 1 and 2. For SEM, provide elementary information with the magnification and voltage used. For figure 2 how authors combined all samples (eg., n=16 samples in one bar plot).
A3: We agree with the reviewer that electron microscopy methods have not been sufficiently detailed. We provided the magnifications (x200, x500 and x835), accelerating voltage (30 kV) and probe current (1 nA) in the figure legend and also added accelerating voltage and probe current into the respective section of Materials and Methods. Further, we added the information on the exact number of samples regarding each plot in Figure 2 into the figure legend.
____________________________________________________________________________
Q4: Figure 3 and 4: box-plot has points but no details which points belongs to which samples more confusing.
A4: We agree with the reviewer that Figures 3 and 4 have not been sufficiently explained. In addition to the ticks and labels indicating the groups at the X axis, we have added the information on which dots belong to which sample groups into the figure legends.
_____________________________________________________________________________
Q5: Figure 5: All samples were pooled? Authors should provide how individual sample has microbial diversity and distributions.
A5: We agree with the reviewer that the number of samples have not been well indicated with regards to Figures 5-8. Hence, we added the number of samples in each group into the figure legends. Indeed, Figure 5 presents the pooled data from all 128 Vent-PM samples. Figure 6 compares the microbial diversity and distributions across surgical (n = 40) and non-surgical (n = 88) units. Figure 7 compares the microbial diversity and distributions across intensive (n = 48) and elective (n = 80) care units. Figure 8 compares the microbial diversity and distributions across health facilities located in coal (n = 27) and chemical (n = 36) districts.
_____________________________________________________________________________
Q6: From the paper: Vent-PM from the surgical units were most frequently contaminated by bacteria 242 (32/40 samples, 80.0%), with only a minor proportion of MDRO (7/40, 17.5% - which 7 samples from the original sample?? So provide separate table do this sample has lots of PM contaminants or not???) and rare 243 (1/40, 2.5%) contamination by viruses or fungi (Figure 6A)
A6: We agree with the reviewer that the contamination of the samples has not been clearly indicated in the manuscript. In Revised Table 4, we provide the detailed information on where each sample has been collected (similar to Revised Table 3) and which pathogens (i.e., bacteria, MDRO, viruses or fungi) contaminated each particular sample. As was suggested by the reviewer, in this table we demarcated overall bacterial contamination and contamination by MDRO.
___________________________________________________________________________
Round 2
Reviewer 2 Report
The Authors have satisfactorily replied to all my questions. I believe the manuscript has been improved and, therefore, I suggest pubblication of manuscript in present form.